# Is There a Difference in Facial Emotion Recognition after Stroke with vs. without Central Facial Paresis?

**DOI:** 10.3390/diagnostics12071721

**Published:** 2022-07-15

**Authors:** Anna-Maria Kuttenreich, Harry von Piekartz, Stefan Heim

**Affiliations:** 1Department of Psychiatry, Psychotherapy and Psychosomatics, Medical Faculty, RWTH Aachen University, Pauwelsstr. 30, 52074 Aachen, Germany; sheim@ukaachen.de; 2Department of Neurology, Medical Faculty, RWTH Aachen University, Pauwelsstr. 30, 52074 Aachen, Germany; 3Department of Otorhinolaryngology, Jena University Hospital, Am Klinikum 1, 07747 Jena, Germany; 4Facial-Nerve-Center Jena, Jena University Hospital, Am Klinikum 1, 07747 Jena, Germany; 5Center of Rare Diseases Jena, Jena University Hospital, Am Klinikum 1, 07747 Jena, Germany; 6Department of Physical Therapy and Rehabilitation Science, Osnabrück University of Applied Sciences, Albrechtstr. 30, 49076 Osnabrück, Germany; h.von-piekartz@hs-osnabrueck.de; 7Institute of Neuroscience and Medicine (INM−1), Forschungszentrum Jülich, Leo-Brand-Str. 5, 52428 Jülich, Germany

**Keywords:** emotion recognition, facial feedback, central facial paresis, stroke

## Abstract

The Facial Feedback Hypothesis (FFH) states that facial emotion recognition is based on the imitation of facial emotional expressions and the processing of physiological feedback. In the light of limited and contradictory evidence, this hypothesis is still being debated. Therefore, in the present study, emotion recognition was tested in patients with central facial paresis after stroke. Performance in facial vs. auditory emotion recognition was assessed in patients with vs. without facial paresis. The accuracy of objective facial emotion recognition was significantly lower in patients with vs. without facial paresis and also in comparison to healthy controls. Moreover, for patients with facial paresis, the accuracy measure for facial emotion recognition was significantly worse than that for auditory emotion recognition. Finally, in patients with facial paresis, the subjective judgements of their own facial emotion recognition abilities differed strongly from their objective performances. This pattern of results demonstrates a specific deficit in facial emotion recognition in central facial paresis and thus provides support for the FFH and points out certain effects of stroke.

## 1. Introduction

Emotion recognition is omnipresent in social interactions [1] and represents an important social competence [2]. Faces provide relevant clues for the recognition of emotions [2,3]. One explanation of the facial recognition of emotions is provided by the Facial Feedback Hypothesis (FFH) [4]. The present study therefore compares stroke patients with vs. without unilateral central facial paresis, i.e., the partial inability to perform facial movements [5], in order to test the FFH prediction of a specific deficit of visual facial emotion recognition in individuals with central facial paresis.

### Emotion Processing and the Role of Facial Feedback

Facial emotion expressions are part of nonverbal communication [3] and are regarded as some of the most important nonverbal features in the identification of emotions [6]. Facial expression can be highly variable due to the precise control of the different facial muscles [1] and their voluntary or affective control [7], although the basic emotions framework considers a set of emotions to be highly elementary, unique and independent of culture, time and place [8]. These basic emotions are: anger, disgust, fear, joy, sadness and surprise [9,10]. Each of the basic emotions is characterized by specific patterns of facial muscle activities [8,11]. These congenital, ubiquitous basic emotions [12] are typically used to observe (facial) emotion recognition [13].

The accuracy of emotion recognition varies, depending on the particular emotion presented. Joy is detected significantly more accurately and quickly than all other basic emotions, whereas fear is detected significantly less accurately and more slowly than the other emotions [14]. The basic emotions of surprise and anger, as well as disgust and sadness, are similarly well-identified in terms of accuracy (performance listed in descending order) [14]. Besides differences per emotion, emotion recognition depends on sex and age. Women are faster at facial emotion recognition than men [15]. With increasing age, emotion recognition performance decreases [16]. It has not yet been conclusively clarified whether the processing of emotions is innate [4,17] or whether a concept of emotions must first be learned [18]. A combination is also conceivable, if basic emotions are considered as biologically anchored [12] and innate [17], while all of the other, more complex emotions [8] have to be learned first [12]. The localisation of emotion processing is also a matter of controversy, with evidence for right, left, or left and right hemispheric activation [19]. Dominance of the right hemisphere has been described historically [20], whereas recent evidence has highlighted a combination of different neuronal networks with different lateralization [19].

In emotion processing, the importance of afferent information from the body is emphasised, e.g., facial expression [18]. In this sense, the FFH provides a theoretical account of the process of facial emotion recognition. It postulates that other persons’ emotions are recognised by one’s own facial information [4]. The decoding requires the imitation of the facial expression of the other person and the corresponding proprioceptive facial feedback [21,22] (‘facial reflex’ is a synonym for ‘facial feedback’ [11]). Neal and Chartrand [22] summarised the working steps of the FFH: (1) imitation of the facial expression of the communication partner (discrete, unconscious, fast, automated and specific to the emotion); (2) transmission of afferent information from the face to the brain; and (3) experience and recognition of the emotion [22].

Whereas a person’s spontaneous, quick and unobtrusive imitation with their own face is basically unproblematic [23], pathological conditions affecting facial integrity may affect the abilities to initiate or imitate basic emotions’ corresponding facial expressions. Such conditions include, for example, facial paresis, a unilateral or bilateral palsy of the facial musculature following a peripheral or central defect [24]. The central form of facial paresis considered in this study typically presents unilaterally, contralateral to the central lesion [25], after stroke [26].

Whether and precisely what role facial feedback plays in emotion recognition has not yet been conclusively clarified. For example, different research results show evidence for and against the FFH in the case of limited facial feedback (due to illness or artificially provoked).

Significant deficits in facial emotion recognition were reported by Konnerth et al. [27] and Storbeck et al. [28] in patients with peripheral facial paresis/paralysis. Konnerth et al. [27] reported that patients achieved lower accuracy values than healthy controls, although the difference was not significant. Storbeck et al. [28] also detected that accuracy in facial emotion recognition did not differ significantly between patients with facial paresis and healthy controls. However, visual emotion recognition was significantly slower compared to the control subjects in both studies [27,28]. More specifically, Korb et al. [29] reported differences depending on the paralysed side of the face, with facial emotion recognition being more affected in patients with left-sided rather than right-sided facial palsy. Such findings might be taken as supportive evidence for the FFH, as persons with intact feedback show faster facial emotion recognition times [22,30,31,32,33]. This reduced accuracy of emotion recognition in patients with peripheral facial palsy could be explained by Niedenthal et al. [33], according to whom self-experienced emotions can be recognized earlier than those that are not self-perceived [33]. In contrast, Keillor et al. [34] did not report differences in the accuracy of emotion naming, discrimination or matching tasks in their single case study of a patient with bilateral facial paralysis in Guillain–Barré syndrome, nor did Bogart and Matsumoto [35] report facial emotion recognition deficits in patients with congenital bilateral facial paresis in Moebius syndrome. However, Calder et al. [36] did observe differences in the accuracy of emotion recognition with respect to at least one basic emotion in patients with Moebius syndrome.

A different way of investigating facial feedback in healthy participants is with an injection of botulinum toxin in the facial muscles for temporarily paralysis. Different studies using this method showed changed emotion recognition in terms of accuracy and time [22,32]. The results may point to a direct link between facial feedback and emotion processing [32].

Besides limited facial movements due to experimental induction and peripheral facial palsy, other disorders could also affect (1) facial movements and (2) facial emotion recognition—for instance, central facial palsy after stroke and Parkinson’s disease. Stroke occurs suddenly due to disturbed blood flow and oxygen deficiency (ischemic stroke) or bleeding (hemorrhagic stroke) in the brain and leads to individual disabilities [37], whereas Parkinson’s disease is a neurodegenerative disorder involving loss of dopamine in the substantia nigra, resulting in typical symptoms of rigor, tremor and bradykinesia [38]. Both central facial palsy after stroke [26,39] and Parkinson’s disease [40,41,42] could result in similar effects, i.e., reduced facial expression and therefore reduced facial feedback. Following the FFH, facial feedback due to facial integrity is needed for facial emotion recognition [23]. Both in stroke [43] and in Parkinson’s disease [41], facial emotion recognition could be impaired. However, there is not necessarily a direct correlation between limitations in facial expression and facial emotion recognition, at least in Parkinson’s disease [41].

In summary, there is evidence that patients with limited facial feedback and facial mimicry abilities (e.g., in peripheral facial paresis) are potentially affected by limited facial emotion recognition. To date, to the best of our knowledge, patients with peripheral facial palsy have been studied, whereas patients with central facial palsy have been overlooked.

The care of patients with central facial palsy is insufficient and rehabilitation guidelines are required [44]. To improve treatment and establish guidelines, deficits or remaining abilities must be identified first. To this end, we designed a study to find proof of facial emotion recognition abilities in patients with central facial palsy.

Consequently, the aim of the study was to test facial emotion recognition in patients with central facial paresis after stroke in terms of accuracy and time with visually presented, i.e., facial, stimuli presented by healthy subjects. Testing different modalities (facial and auditory) in two patient groups (with or without facial paresis after stroke) allows assessment of whether there is a general deficit in emotion recognition—which is a possibility after stroke [43]—or whether only one particular modality is (more) affected. If there are no deficits in emotion recognition at all, i.e., if the performance is comparable to that of healthy control subjects, it can assume that emotion recognition may be intact. Accordingly, the primary research question was: Can patients with central facial paresis after stroke recognise facial emotions?

## 2. Materials and Methods

### 2.1. Participants

Three groups of participants were considered for this study: (1) patients with unilateral central facial paresis after stroke, (2) patients without facial paresis after stroke and (3) healthy subjects. The data for the patient groups (1) and (2) were collected within the study (data are available from the authors on request), whereas the reference values for the healthy subject group (3) were already available [45,46,47] and served for an additional comparison.

The inclusion and exclusion criteria are summarised in Table 1. The patients were referred by various cooperation partners, hospitals and local practices for speech–language therapy. Recruitment and data collection took place in the period from 22 February until 14 May 2019 in Germany.

A total of 67 patients were recruited. Four of these were drop-out cases (one case: disorientation; one case: suspected bucco–facial apraxia with no possibility of assessing facial paresis; two cases: antidepressant medication with suspected altered emotional regulation). The remaining 63 patients were assigned to the study group (patients with central facial paresis, n = 34) or the control group (patients without facial paresis, n = 29) according to their diagnosis of facial paresis. Sociodemographic data and information on lesions, facial paresis, general mental capacities and aphasia for the study and control groups are given in Table A1, Table A2, Table A3, Table A5 and Table A6 (Appendix A).

The study was approved by the local ethics committee (key: EK 271/18) of the Medical Faculty at RWTH Aachen University, and all regulations of the ethics committee were implemented. All experiments were performed in accordance with the relevant guidelines and regulations. All participants signed an informed consent form after receiving detailed information.

### 2.2. Materials

For both facial emotion recognition and auditory emotion recognition, the same conditions were set, i.e., an item was presented (visually or auditorily) and the patients had ten seconds to respond. There were different options available as answers. The respective software systems recorded accuracy and time. For both modalities, a pre-test with ten items (initially randomized, later presented in the same order) was performed. The pre-test ensured that the task was understood [48] (see, also, Appendix B).

#### 2.2.1. Visual Facial Emotion Recognition

In our study, we opted to use the *Myfacetraining* (MFT) Program (CRAFTA Cranio Facial Therapy Academy, Hamburg, Germany) [47,49], which consists of a standard test for accuracy and time taken for facial emotion recognition [47,49]. Forty-two subjects, each showing a basic emotion with their face, were presented on a screen. The person was first shown in a neutral position before changing to an emotional facial expression (basic emotion). Six additional answer options were displayed on the screen according to the basic emotion [47] (see, also, Appendix B).

#### 2.2.2. Auditory Emotion Recognition

In addition to faces, voices (auditory) are the most important modalities in emotional communication [1]. A sub-portion of the *Montreal Affective Voices (MAVs)* [45] was used for the assessment. These are emotional, non-linguistic, vocal expressions of /a/ (to be compared with *a* as in *apple*, British English). Sixty items for the six basic emotions [45] were used. The *Montreal Affective Voices* were presented with a specially programmed experiment with the software *PsychoPy*, version 3.0.0b9 [50] (see, also, Appendix B).

#### 2.2.3. Subjective Facial Emotion Recognition: Self-Assessment Questionnaires, Emotion Recognition

Coulson et al. [51] asked relatives of patients with facial paresis for their assessments of emotional recognition. Based on this, two standardized questionnaires were designed for the present study which enabled the systematic collection of subjective facial emotion recognition data. The *Self-Assessment Questionnaires Emotion Recognition Accuracy* and *Time* were used to document self-assessment of facial emotion recognition of the six basic emotions (anger, disgust, fear, joy, sadness and surprise) [51]. In order to be able to look at the evaluation in a differentiated way, one questionnaire was developed to assess accuracy and another was developed to assess time taken for facial emotion recognition. The questionnaires assess possible changes between pre-morbid and current abilities per basic emotion. The questions that featured in the questionnaires in each case were as follows: *How well do you recognize the following feelings in other people’s faces?* One of three answer options could be selected for each questionnaire. For *Accuracy*, the patient evaluated whether the basic emotion in question was *more difficult*, *just as well as* or *more easily* recognised than before stroke. For *Time*, the patient indicated whether the basic emotion was detected *slower*, *as fast as* or *faster* than before stroke. For deteriorations (indicated by the response options *more difficult* or *slower*), a score of −1 was assigned. If the patient did not notice any changes (response options *just as well as* or *just as fast as*), zero points (0) were recorded. For improvements (answer options *easier* or *faster*), the patient achieved a score of +1, resulting in a score between −6 and +6 per questionnaire.

#### 2.2.4. Sunnybrook Facial Grading System for Diagnosing Facial Paresis

In order to answer the main research question, all patients were examined in a standardised way to identify possible facial paresis. Only this allowed to divide the patients into the study group (participants with central facial paresis) or the control group (participants without central facial paresis). The *Sunnybrook Facial Grading System* [52,53] is used for the standardised assessment for diagnosing facial paresis or paralysis, respectively. This measurement method is explicitly recommended [54]. It is also considered the current standard in the evaluation of facial paresis [55] and has been used in various studies (e.g., [54,56,57,58,59,60,61,62]). Ross et al. [52] published the original version of the *Sunnybrook Facial Grading System* in 1996, which was implemented in the present study (German version [53]). For this purpose, a video was made of each patient with an *Apple iPod touch* (camera at right angles, at the individual height of the chewing plane, 150 cm from the patient’s chin), in which the patients were asked in a standardised manner to show their face at rest or to perform an arbitrary movement with their face (raise eyebrows, close eyes gently, smile with open mouth, show teeth, pucker lips). The videos were evaluated by a speech–language therapist (see, also, Appendix B).

### 2.3. Statistical Analysis

Two-factorial ANOVAs with post-hoc *t*-tests were performed with the factors *group* (with vs. without facial paresis) as between-subject factor and *modality* (facial vs. auditory emotion recognition) as within-subject factor. Accuracy and time taken for emotion recognition were considered as dependent variables. In order to compare the empirical data obtained in the present study with normative data for healthy controls (without stroke and without facial paresis) which were already available, a series of *t*-tests were subsequently performed separately both for accuracy and time. To compare facial emotion recognition and auditory emotion recognition in terms of accuracy and time in patients and healthy subjects, *t*-tests were performed for one sample. For the comparison between patients with and without facial paresis, two-factorial ANOVAs and (post-hoc) *t*-tests for independent samples were run. *t*-tests for dependent samples were performed to compare facial emotion recognition and auditory emotion recognition in patients with and without facial paresis. To analyse subjective emotion recognition in terms of accuracy and time, one-sample *t*-tests were conducted. To compare accuracy and time, *t*-tests for dependent samples were performed.

Benjamini–Hochberg correction was applied if more than one *t*-test was conducted.

## 3. Results

The results for objective (accuracy and time) and subjectively perceived success in emotion recognition are summarised in Figure 1, Figure 2, Figure 3, Figure 4 and Table A4 (Appendix A).

### 3.1. Accuracy of Facial Emotion Recognition 

The results of the ANOVA for accuracy were a main effect of *group* F(1;61) = 6.620; *p* = 0.013, a main effect of *modality* F(1;61) = 96.535; *p* < 0.001 and an interaction effect *group x modality* F(1;61) = 18.330; *p* < 0.001, which means that participants with central facial paresis recognised visually presented basic emotions significantly worse (reduced accuracy) compared to participants without facial paresis (t(49.425) = −3.767; *p* < 0.001; after correction p = 0.002) and compared to healthy controls (t(33) = −22.888; *p* < 0.001; after correction *p* = 0.002). Participants without facial paresis recognised visually presented basic emotions significantly worse (reduced accuracy) compared to healthy controls (t(28) = −10.476; *p* < 0.001; after correction *p* = 0.002), Figure 1.

### 3.2. Accuracy of Auditory Emotion Recognition

Participants with central facial paresis recognised auditorily presented basic emotions significantly worse (reduced accuracy) compared to healthy controls (t(33) = −13.258; *p* < 0.001; after correction *p* = 0.002). Participants without facial paresis recognised auditorily presented basic emotions significantly worse (reduced accuracy) compared to healthy controls (t(28) = −11.259; *p* < 0.001; after correction *p* = 0.002). Participants with vs. without central facial paresis did not differ significantly in auditory emotion recognition (accuracy) (t(61) = 0.616; *p* = 0.540; after correction *p* = 0.540), Figure 2.

### 3.3. Comparison of Accuracy of Facial and Auditory Emotion Recognition

Participants with central facial paresis recognised visually presented basic emotions significantly worse (reduced accuracy) than auditorily presented basic emotions (t(33) = −11.252; *p* < 0.001; after correction *p* = 0.002). Participants without facial paresis recognised visually presented basic emotions significantly worse (reduced accuracy) than auditorily presented basic emotions (t(28) = −3.485; *p* = 0.002; after correction *p* = 0.002).

### 3.4. Time Taken for Facial Emotion Recognition

The results of the ANOVA for accuracy were a main effect of *group* (F(1;61) = 2.797; *p* = 0.100), a main effect of *modality* (F(1;61) = 3.311; *p* = 0.074), and an interaction effect *group × modality* (F(1;61) = 3.148; *p* = 0.081)), which means that participants with central facial paresis did not recognise visually presented basic emotions significantly more slowly (reduced time) compared to participants without facial paresis (t(61) = 0.414; *p* = 0.680; after correction *p* = 0.680). Participants with central facial paresis recognised visually presented basic emotions significantly (not significantly after correction) faster (increased time) compared to healthy controls (t(33) = −2.442; *p* = 0.020; after correction *p* = 0.060). Participants without facial paresis recognised visually presented basic emotions significantly faster (increased time) compared to healthy controls (t(28) = −2.390; *p* = 0.024; after correction *p* = 0.036), Figure 3.

### 3.5. Time Taken for Auditory Emotion Recognition

Participants with vs. without central facial paresis did not differ significantly with respect to the average time taken for auditory emotion recognition (t(61) = −1.851; *p* = 0.069), Figure 4.

### 3.6. Comparison of Time Taken for Facial and Auditory Emotion Recognition

Participants with central facial paresis recognised visually presented basic emotions significantly (not significantly after correction) faster (increased time) than auditorily presented basic emotions (t(33) = −2.269; *p* = 0.030; after correction *p* = 0.060). Participants without facial paresis recognised visually presented basic emotions not significantly differently to auditorily presented basic emotions (t(28) = −0.041; *p* = 0.968; after correction *p* = 0.968).

### 3.7. Subjective Judgement of Emotion Recognition from the Perspective of Participants with Central Facial Paresis

Both the average accuracy of facial emotion recognition (mean = −0.71 ± 1.90) was perceived as significantly limited (t(33) = −2.167; *p* = 0.038; after correction *p* = 0.038) and the time taken for facial emotion recognition (mean = −1.91 ± 2.90) was subjectively perceived as significantly limited (t(33) = −3.849; *p* = 0.001; after correction *p* = 0.003) in participants with central facial paresis. Participants with central facial paresis judged themselves to be significantly more restricted in terms of the time taken for facial emotion recognition than in terms of accuracy (t(33) = 2.689; *p* = 0.011; after correction *p* = 0.017), Figure 5.

### 3.8. Further Analysis

In order to verify that the identified pattern is reasonable on the basis of these results, the following further control calculations were made.

A correlation calculation (Pearson’s product moment correlation) between objective accuracy and objective time taken for facial emotion recognition in patients with and without central facial paresis was performed. The accuracy of and the time taken for facial emotion recognition in patients with central facial paresis were positively correlated with each other (r = 0.729; *p* < 0.001). The average accuracy and the average time taken for facial emotion recognition in patients without facial paresis were not significantly correlated with each other (r = 0.291; *p* = 0.126).

Furthermore, a correlation calculation (Pearson’s product moment correlation) between objective facial emotion recognition, accuracy and severity of facial paresis using the Sunnybrook Facial Grading System across all patients (with and without facial paresis) was performed. The average accuracy of facial emotion recognition and the severity of facial paresis were significantly positively correlated with each other (r = 0.31; *p* = 0.014).

Moreover, a one-tailed *t*-test on independent samples for facial emotion recognition accuracy showed no significant difference between patients with left-sided facial paresis (mean = 26.44 ± 11.49) and right-sided facial paresis (mean = 29.25 ± 10.69), t(32) = −0.734; *p* = 0.234). Another one tailed *t*-test on independent samples for facial emotion recognition time showed no significant difference between patients with left-sided facial paresis (mean = 3.12 ± 0.48) and right-sided facial paresis (mean = 3.17 ± 0.47), t(32) = −0.322; *p* = 0.375.

Furthermore, a chi-squared test to compare the number of patients with limitations in general mental capacity in both groups (Table A5, Appendix A) was performed. Both groups were comparable, with x^2^(1, n = 63) = 0.204; *p* = 0.651. Another chi-squared test to compare the number of patients with aphasia in both groups (Table A6, Appendix A) was also carried out. Both groups were comparable, with x^2^(1, n = 63) = 1.546; *p* = 0.214.

Additionally, univariate and multivariate regressions, with emotion recognition (facial and auditory, accuracy and time taken) as the dependent variable and predictors diagnosis of facial paresis, sex, age, subjective judgement, general mental capacity and time post-onset as independent variables, were conducted (Table A7 and Table A8, Appendix A). Patients with facial paresis recognised visually presented basic emotions significantly worse (reduced accuracy) compared to patients without facial paresis, as calculated by means of univariate regression (beta = −0.444; *p* < 0.001) as well as by multivariate regression (beta = −0.353; 0.003).

## 4. Discussion

This study investigated visual facial emotion recognition (VFER) in patients with and without central facial paresis vs. healthy individuals. The results of our study showed that the participants with central facial paresis had significantly lower average accurate emotion recognition abilities with respect to the facial modality compared to the auditory modality. The less accurate VFER in cases of facial paresis but not in auditory emotion recognition may be due to changes in the facial feedback mechanism. Clinically, this means that VFER in persons with limited facial mimicry abilities, as in central facial paresis patients, does appear to be affected, in contrast to auditory recognition [36]. Taking into account that we did not test facial mimicry itself (i.e., facial muscle activity was not measured during the emotion-recognition task), but facial emotion recognition, facial paresis can be inferred to be one factor influencing the accuracy of objective facial emotion recognition, which may be affected by changes in the facial feedback mechanism. This may be an indication that the accuracy of objective facial emotion recognition is especially limited when facial feedback is altered by facial paresis. Auditory performance does not appear to be affected by facial paresis (for a similar finding, cf. [36]). Besides facial paresis, stroke, also, could be one factor influencing the accuracy of objective facial emotion recognition in our sample. All participants (with and without facial paresis) had had at least one stroke. Since stroke may also cause deficits in emotion recognition [43], our examined patient groups may be affected as well. These two potential factors (altered facial feedback and altered central processing due to stroke) indicate the relevance of and need to study patients without stroke but with limited facial feedback—for example, patients with peripheral facial palsy.

Our results reveal significant deficits in terms of accuracy of facial emotion recognition, in contrast with other studies that did not report any differences, e.g., [27,28,34]. This fact may be due to the large sample size (participants with facial paresis: n = 34; participants without facial paresis: n = 29) and the inclusion of different phases post-onset, with a wide range since the time of stroke (day 5 up to day 6361 post-onset). However, previous studies reported significant limitations in terms of average time taken for facial emotion recognition, e.g., [27,28], while the participants in the present study showed faster reaction times. This, in turn, could indicate that the participants after stroke replied *quick and dirty* [63], while they suffered from other impairments, such as deficits in attention, concentration and memory [64], in addition to the facial paresis after stroke. In order to investigate a possible systemic connection between the fast, inaccurate responses, the significant positive correlation between the objective accuracy and the objective time taken for facial emotion recognition in patients with facial paresis provides further insight: the faster a patient with facial paresis responded, the less accurate was the response, whereas no correlation was found in patients without facial paresis. This could indicate that the patients with facial paresis were themselves aware of their deficit in the time taken for facial emotion recognition (as reported in the *Self-Assessment Questionnaires Emotion Recognition*) but wanted to show their best performance in the test situation and therefore answered as quickly as possible.

The participants with facial paresis subjectively felt limited both in terms of parameter accuracy and time in VFER. They stated that they were more impaired with respect to time than accuracy. The participants felt that facial emotion recognition had slowed down considerably since the stroke and was somewhat less accurate. These results provide a new insight into subjective emotion recognition, as this was not considered in previous studies. However, the clinical measurement gave contradictory results and showed that the patients were clearly less accurate but faster. Thus, the measured performance appears to be controversial to the subjectively perceived performance.

In the present study, we considered the difference in facial and auditory emotion recognition shown in the results. This may support, for example, FFH, as mentioned before. Nevertheless, it should be noted that a large part of human emotion is communicated via the face *and* the voice, as discussed in the literature. To the best of our knowledge, this is the first clinical study which combines two different modalities in a clinical setting [65]. The mentioned factors (limitations such as deficits in attention, concentration and memory [64], besides facial paresis and emotion recognition) influenced both the study results and everyday communication in the patient groups. Although for stroke patients their survival is of primary importance [66], participation is also highly relevant, particularly in the post-acute and chronic phase [67]. Since both groups of patients showed a significant reduction in the accuracy of facial and auditory emotion recognition compared to healthy subjects, intervention recommendations for both groups and both modalities are required. Although there is limited evidence for FFH [68], FHH can be used as an explanation for assessment and rehabilitation [69].

### 4.1. The Relevance of Assessment of Emotion Recognition

The described results not only provide evidence for the FFH and certain effects of stroke but also have implications for the treatment of patients with central facial paresis after stroke. As early as 2013, Dobel et al. [69] called for the examination of facial emotion recognition in patients with facial paresis using basic emotions [69]. In summary, the present study supports this demand and once again advocates it.

Since the accuracy of facial emotion recognition can be impaired, especially in patients with facial paresis after stroke, appropriate assessment and therapy is recommended for this patient group. Deficits should be assessed because the performance limitations may have negative consequences on communication and may increase over time. If the performance of emotion recognition remains impaired, this can lead to the development of disorders such as alexithymia (the inability to recognise or describe one’s own emotions) [11,70]. For example, if sadness is not adequately interpreted, a patient may react defensively and thus not appropriately to a situation [6]. The effects of facial emotion recognition are therefore far-reaching and decisive for adequate social contact. The somewhat controversial results for the objective measurement and subjective assessment of facial emotion recognition in participants with facial paresis require detailed and individual examination in clinical practice. It is not sufficient to *either* ask the patient for his or her opinion *or* carry out an objective diagnosis. Both options should be taken and the results should be compared.

In addition, the lack of disease insight to be expected according to the available results (comparison between clinical measurement and subjective assessment) must become a focus of treatment in order to show the patient the relevance of facial emotion recognition therapy. This should not underestimate the importance of considering the individual wishes and goals of the patient and including them in the sense of joint decision making [71]. The basis for this is the tripartite evidence-based practice [71,72]. This ensures not only the effectiveness and efficiency of therapy, but also therapy motivation and transfer into the patient’s everyday life [71].

### 4.2. Limitations of the Study

The composition of the sample may be considered a limiting factor of the study. A larger and more representative, homogeneous sample tested at the same time post-onset after stroke and subdivided according to the subtypes of central facial paresis (voluntary and involuntary central facial paresis [73]) would therefore be desirable for future studies. For a more precise observation of the lesion localization and comparability of patients, imaging with detailed description of affected brain areas would be useful. In addition, statistical adjustment for different stroke locations and lesion sizes would be beneficial, as differences in emotion recognition could depend on the hemisphere affected [43]. Despite the possibility of different lesion locations and lesion sizes, the results for facial emotion recognition showed significant differences between the patient groups. Since significant effects can already be observed in our sample, we expect similar or stronger effects to be observed with more carefully selected samples with stricter inclusion criteria in further studies. Furthermore, a strong and reliable test battery to assess cognitive capacity (see [74]) is needed to differentiate deficits in emotion recognition and limitations on general mental capacity after stroke. Since emotion perception depends on general mental capacity [74,75,76], any emotion perception test measures general mental capacity to some degree. In the present study, there were comparable numbers of patients with limitations in mental capacity and aphasia, as proven by chi-squared tests. In future studies, comparability should be extended and improved by standardised diagnostics.

However, the significant positive correlation observed between objective facial emotion recognition accuracy and severity of facial paresis, calculated using the *Sunnybrook Facial Grading System* across all patients, points to facial paresis as the main differentiator between the two patient groups. Thus, the higher the accuracy of facial emotion recognition, the higher the score on the *Sunnybrook Facial Grading System*. That is, facial competence correlates with accuracy in facial emotion recognition, or the lower the facial competence, the worse the accuracy in facial emotion recognition. Moreover, significant univariate and multivariate regressions documented the relation between facial emotion recognition accuracy and facial paresis. These results demonstrate the influence of facial paresis on facial emotion recognition once more, but only in terms of accuracy. No significant differences were detected with respect to objective facial emotion recognition accuracy and time taken between patients with left- or right-sided facial paresis. If one hemisphere is dominant in emotion processing [43], patients with lesions in this dominant hemisphere with contralateral facial paresis [25] could possibly be more affected. We cannot confirm this hypothesis and previous research on facial palsy that reported that patients with left-sided facial palsy showed lower performance in facial emotion recognition compared with patients with right-sided facial palsy [29]. However, our results are in line with findings for patients with Parkinson’s disease, where facial asymmetry is not related to hemispheric dominance for emotion processing [77]. Further evidence is needed, then, to inspect possible differences in facial emotion recognition and expression depending on the side affected with facial palsy and on hemisphere.

Perfect comparability of the standard data with the sample data cannot be guaranteed without gaps—for instance, due to the age of the participants (e.g., the *Montreal Affective Voices* validation sample with an average age of 23.3 ± 3 years [45] vs. the patients with facial paresis with an average age of 62.6 ± 9.3 years and patients without facial paresis average with an average age of 58.4 ± 10.7 years). It must also be noted that only a small sample size of normative data (n = 29) was used for the auditory emotion recognition assessment *(Montreal Affective Voices)* [45]. Furthermore, the measurement of auditory and facial emotion recognition is not completely comparable. Especially with regard to the time taken for emotion recognition, it should be noted, for example, that the response modes differed (selecting an option on screen vs. pointing to a surface) and that the numbers of items and response options were not identical. As a consequence, for further research, normative data from healthy individuals should be freshly collected, with comparability extended to the patient groups. Moreover, measurement in facial and auditory emotion recognition tasks should be made even more comparable.

The separate presentation of facial and auditory items in emotion recognition assessments should also be critically questioned. Facial and auditory expressions are not necessarily independent as they can mutually influence their recognition. For example, a facial expression can be generated by moving the mouth while a vocal expression is also made [1]. However, a separation of the modalities, i.e., just visual or just auditory impressions, seemed to make sense in this study in order to differentiate and compare performances. In order to be able to answer the question reliably, this seems unavoidable. At the same time, however, this separate type of emotion recognition is far removed from everyday life and thus reduces the external validity. Equally adapted to optimal experimental conditions, static photographs instead of everyday situations were used [78]. A person is able to show up to 8000 different emotional facial expressions with his or her face [17]. However, it should be critically noted that our study only examined emotion recognition with respect to basic emotions and thus minimized the requirements compared to non-verbal communication in everyday life. It should be noted here that basic emotions can be regarded as the basis for far more complex emotions or emotional states [8]. However, since the recognition of the comparatively primitive basic emotions [8] was assessed as limited in the present study, an even worse performance can be expected for more complex emotions.

## 5. Conclusions

From this study, it may be concluded that:-After a stroke, participants with central facial paresis were significantly less accurate in visually recognising basic emotions compared with stroke patients without facial paresis and compared with a sample of healthy controls;-Auditory emotion recognition in both stroke groups was less accurate than in the control sample;-The facial emotion recognition accuracy of participants with central facial paresis was significantly worse than the auditory accuracy of emotion recognition;-Since visual emotion recognition was clearly worse than auditory emotion recognition in participants with facial paresis after stroke, facial mimicry probably plays an important role in communication with patients after stroke;-The results of our observational study may indicate the overall effects of stroke on emotion recognition and support the FFH, which is a practical and appropriate model implemented in clinical assessments and interventions;-Future research should investigate patients with facial palsy without stroke to further explore the impact of facial feedback on emotion recognition.

## Figures and Tables

**Figure 1 diagnostics-12-01721-f001:**
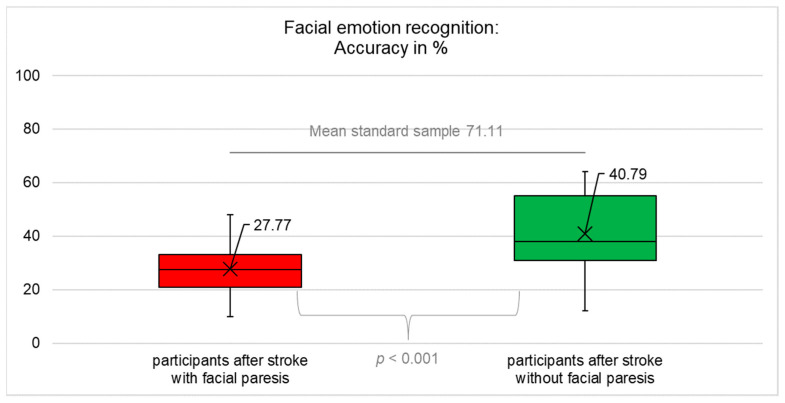
Accuracy of facial emotion recognition (mean, median, interquartile range). Participants after stroke with facial paresis performed significantly worse compared to healthy controls (*p* < 0.001) and compared to participants after stroke without facial paresis (*p* < 0.001). The data for healthy controls were not collected in this study but were taken from [46,47], so no information on the actual distribution of the data is available but only the mean as an indicator of the central tendency. Therefore, the figures only contain two box plots, not three.

**Figure 2 diagnostics-12-01721-f002:**
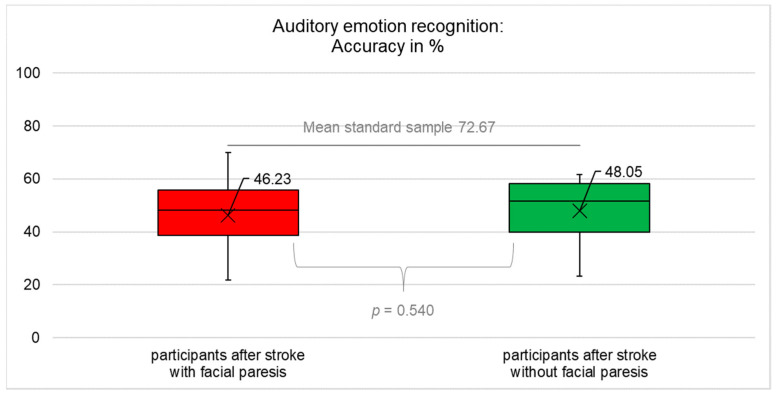
Accuracy of auditory emotion recognition (mean, median, interquartile range). Participants after stroke with facial paresis performed significantly worse compared to healthy controls (*p* < 0.001) but did not differ significantly compared to participants after stroke without facial paresis (*p* = 0.540). The data for healthy controls were not collected in this study but were taken from [45], so no information on the actual distribution of the data is available but only the mean as an indicator of the central tendency. Therefore, the figures only contain two box plots, not three.

**Figure 3 diagnostics-12-01721-f003:**
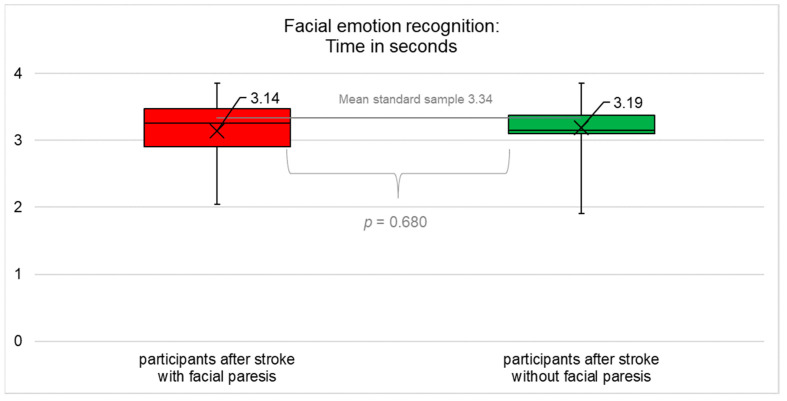
Average time of facial emotion recognition (mean, median, interquartile range). Participants after stroke with facial paresis performed significantly faster compared to healthy controls (*p* = 0.02) but did not differ significantly compared to participants after stroke without facial paresis (*p* = 0.68). The data for healthy controls were not collected in this study but were taken from [46,47], so no information on the actual distribution of the data is available but only the mean as an indicator of the central tendency. Therefore, the figures only contain two box plots, not three.

**Figure 4 diagnostics-12-01721-f004:**
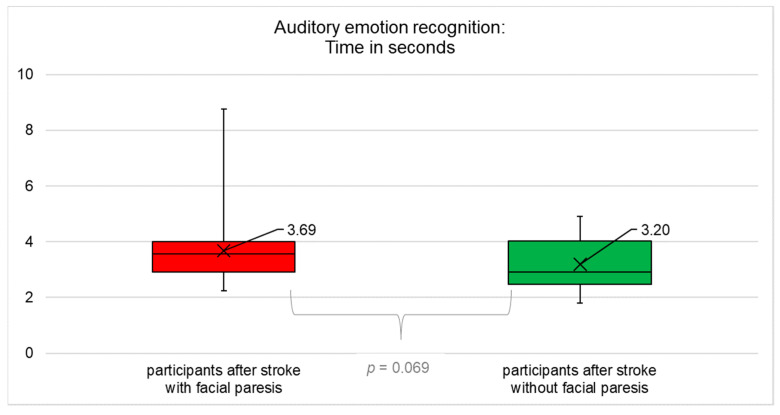
Average time taken for auditory emotion recognition (mean, median, interquartile range). Participants after stroke with facial paresis did not differ significantly compared to participants after stroke without facial paresis (*p* = 0.069).

**Figure 5 diagnostics-12-01721-f005:**
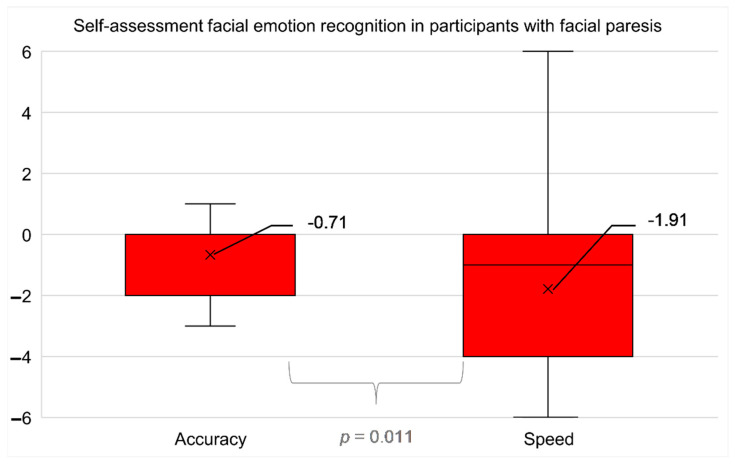
Accuracy and time taken in subjective facial emotion recognition (mean, median, interquartile range) in participants after stroke with facial paresis. Participants felt significantly more restricted in terms of time compared to accuracy (*p* = 0.011).

**Table 1 diagnostics-12-01721-t001:** Inclusion and exclusion criteria.

Inclusion Criteria	Exclusion Criteria
Adult persons (≥18 years) with or without unilateral central facial paresis after stroke (ischemic or hemorrhagic)	Children and adults with peripheral facial paresis
Acute, post-acute or chronic phase of stroke	Other neurological or psychological diseases
For the investigation: - Capacity for approximately 75 min, sitting for approximately 10 min - Ability to choose answer options - Communication skills needed to follow instructions and to answer questionnaires	For the investigation: - Impairment of general status, communication skills and/or ability to answer such that the investigation would not be possible
Normal or corrected visual and hearing ability	
Ability to consent	No ability to consent

## Data Availability

The data presented in this study are available on request from the corresponding author. The data are not publicly available due to their having been collected as part of a larger research project that has not yet been completed.

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
