# Peer review of "Is There a Difference in Facial Emotion Recognition after Stroke with vs. without Central Facial Paresis?"

_diagnostics, 2022, doi:10.3390/diagnostics12071721_

Round 1
Reviewer 1 Report
The shortcommings regarding the study design remain the same. Furthermore, an experimental design requires its own control group. It is not sppropriate to refere to reference values reported in literature.
Author Response
Response to Reviewer 1 Comments
Point 1: The shortcommings regarding the study design remain the same. Furthermore, an experimental design requires its own control group. It is not appropriate to refere to reference values reported in literature.
Response 1: Thank you very much for your review.
In addition to existing norm data from the literature, we examined both a control group and a control condition. The control group were patients after stroke without facial palsy. The control condition was a task of auditory emotion recognition.
We described this approach in the manuscript, section Introduction, lines 135-145:
“Consequently, the aim of the study was to test facial emotion recognition of patients with central facial paresis after stroke in accuracy and time with visually presented, i.e. facial stimuli, in healthy subjects. Testing different modalities (facial and auditory) in two patient groups (with or without facial paresis after stroke) allows assessing whether there is a general deficit in emotion recognition
– this could be possible after stroke [43] – or whether only one particular modality is (more) affected.
If there are no deficits in emotion recognition at all, i.e. if the performance is comparable to that of healthy control subjects, it may be assumed that emotion recognition may be intact. Accordingly, the primary research question was: Can patients with central facial paresis after stroke recognise facial emotions?”

Reviewer 2 Report
I would like to thank the authors for their responses and revisions. I agree with them that this study is the first to investigate the facial feedback hypothesis in such samples with this power. Nevertheless, I still believe a more multivariate design (including actual general mental ability measured) would have been necessary to really test the facial feedback hypothesis. However, the authors did a great job in replying to my comments and adjusting the manuscript accordingly. I highly appreciate the extended limitations section and the addition of information regarding general mental deficits of participants. This helps to somewhat answer one of my points of critique. Although I still believe that a stronger design would have been better here, I cannot see major flaws in the manuscript anymore and believe this manuscript could be accepted.
Author Response
Response to Reviewer 2 Comments
Point 1: I would like to thank the authors for their responses and revisions. I agree with them that this study is the first to investigate the facial feedback hypothesis in such samples with this power. Nevertheless, I still believe a more multivariate design (including actual general mental ability measured) would have been necessary to really test the facial feedback hypothesis. However, the
authors did a great job in replying to my comments and adjusting the manuscript accordingly. I highly appreciate the extended limitations section and the addition of information regarding general mental deficits of participants. This helps to somewhat answer one of my points of critique. Although I still believe that a stronger design would have been better here, I cannot see major flaws in the manuscript anymore and believe this manuscript could be accepted.
Response 1: Thank you very much for your renewed review on the revised manuscript.
We are happy to be able to answer your points of critique and improve the manuscript with your review.
In future research, a stronger design should be applied, as already suggested by you and incorporated in the manuscript, section Discussion, Limitations of the study, lines 456-523, espacially for general mental deficits , lines 468-475.

Reviewer 3 Report
In the present study, auditory and emotion recognition was tested in patients with central facial paresis, in patients without facial paresis, both after stroke, and in healthy controls. Accuracy for emotion recognition was significantly worse in patients with vs. without facial paresis and also in comparison to healthy controls. For patients with facial paresis, the accuracy for facial emotion recognition was significantly worse than for auditory emotion recognition. Finally, in patients with facial paresis, the subjective judgements of their own facial emotion recognition ability differed strongly from their objective performance. This pattern of results demonstrates a specific deficit for facial emotion recognition in central facial paresis.
The manuscript is of interest and well written. I do have only a few comments:
- Title: I would suggest to remove the last part after the question mark (e.g., "An observation study"), or perhaps change with "Results from a single-center study" (or something similar). However, I do not believe here the term "observation study" would be appropriate.
- Regarding the manuscript I would suggest to give a wider description in the introduction of "emotion recognition" and "emotion expression." Also, a brief description of the disorders in which emotion recognition could be affected, would be of great help (e.g., hemispheric lesions versus parkinsonism) (For further references see also: Bologna M, et al. Altered Kinematics of Facial Emotion Expression and Emotion Recognition Deficits Are Unrelated in Parkinson's Disease. Front Neurol. 2016 Dec 14;7:230. doi: 10.3389/fneur.2016.00230; Bologna M, et al. Facial bradykinesia. J Neurol Neurosurg Psychiatry. 2013 Jun;84(6):681-5. doi: 10.1136/jnnp-2012-303993; and Marsili L, et al. Bradykinesia of posed smiling and voluntary movement of the lower face in Parkinson's disease. Parkinsonism Relat Disord. 2014 Apr;20(4):370-5. doi: 10.1016/j.parkreldis.2014.01.013. Epub 2014 Jan 22. PMID: 24508573).
- Regarding the study per se, I believe the results here reported are of great interest and maybe a comment in the discussion on the sidedness of the lesion (e..g., right vs left) and on its possible effect on both emotion expression and recognition would be of great interest. Did the authors investigate possible differences according to right or left facial palsy? Please, argue (See also: Ricciardi L, et al. Emotional facedness in Parkinson's disease. J Neural Transm (Vienna). 2018 Dec;125(12):1819-1827. doi: 10.1007/s00702-018-1945-6; and Marsili L, et al. Unraveling the asymmetry of Mona Lisa smile. Cortex. 2019 Nov;120:607-610. doi: 10.1016/j.cortex.2019.03.020).
Author Response
Response to Reviewer 3 Comments
In the present study, auditory and emotion recognition was tested in patients with central facial paresis, in patients without facial paresis, both after stroke, and in healthy controls. Accuracy for emotion recognition was significantly worse in patients with vs. without facial paresis and also in comparison to healthy controls. For patients with facial paresis, the accuracy for facial emotion
recognition was significantly worse than for auditory emotion recognition. Finally, in patients with facial paresis, the subjective judgements of their own facial emotion recognition ability differed strongly from their objective performance. This pattern of results demonstrates a specific deficit for
facial emotion recognition in central facial paresis.
The manuscript is of interest and well written. I do have only a few comments:
Response: Thank you very much for your review and constructive feedback.
We addressed all your points in the reviesed manuscript and listed detailed explanation below.
Point 1: Title: I would suggest to remove the last part after the question mark (e.g., "An observation study"), or perhaps change with "Results from a single-center study" (or something similar). However, I do not believe here the term "observation study" would be appropriate.
Response 1: Thank you for your porposed amendment.
We removed the last part after the question mark “An observation study”. The new titel of the manuscript is now “Is there a difference in Facial Emotion Recognition after Stroke with vs. without Central Facial Paresis?”, line 3.
Point 2: Regarding the manuscript I would suggest to give a wider description in the introduction of "emotion recognition" and "emotion expression." Also, a brief description of the disorders in which emotion recognition could be affected, would be of great help (e.g., hemispheric lesions versus parkinsonism) (For further references see also: Bologna M, et al. Altered Kinematics of Facial Emotion Expression and Emotion Recognition Deficits Are Unrelated in Parkinson's Disease. Front Neurol. 2016 Dec 14;7:230. doi: 10.3389/fneur.2016.00230; Bologna M, et al. Facial bradykinesia. J Neurol
Neurosurg Psychiatry. 2013 Jun;84(6):681-5. doi: 10.1136/jnnp-2012-303993; and Marsili L, et al. Bradykinesia of posed smiling and voluntary movement of the lower face in Parkinson's disease. Parkinsonism Relat Disord. 2014 Apr;20(4):370-5. doi: 10.1016/j.parkreldis.2014.01.013. Epub 2014 Jan
22. PMID: 24508573).
Response 2: Thank you for your comments.
We expanded the description of emotion expression in the Introduction, lines 42-45:
“Facial emotion expressions are part of nonverbal communication [3] and regarded as one of the most important nonverbal features in the identification of emotions [6]. Facial expression could be highly variable due to the precise control of the different facial muscles [1] and their voluntary or affective
control [7].”
And also we expanded the description of physiological emotion recognition in the Introduction, lines 51-65:
“The accuracy of emotion recognition varies, depending on the particular emotion presented. Joy is detected significantly more accurately and quickly than all other basic emotions, whereas fear is detected significantly less accurately and more slowly than the other emotions [14]. The basic emotions surprise and anger as well as disgust and sadness are similarly well-identified in accuracy (performance listed in descending order) [14]. Besides differences per emotion, emotion recognition depends on sex and age. Women are faster in facial emotion recognition than men [15]. With increasing age, emotion recognition performance decreases [16]. It has not yet been conclusively
clarified whether the processing of emotions is innate [4] [17] or whether a concept of emotions must first be learned [18]. A combination is also conceivable, if basic emotions are considered biologically anchored [12] and innate [17], while all other, more complex emotions [8] have to be learned first [12]. The localisation of emotion processing is also controversy discussed with evidence for right, left, or left and right hemispheric activation [19]. Dominance of the right hemisphere has been described historically [20], whereas recent evidence highlighted a combination of different neuronal networks with different lateralisation [19].”
Also, we integrated a description of other disorders (as recommended stroke vs. Parkinson’s Disesase) affecting emotion expression and recognition in the Introduction, lines 113-126 and added the recommended literature.
“Besides provoked limited facial movements, or limited facial movements due to peripheral facial palsy, also other disorders could affect (1) facial movements, and (2) facial emotion recognition, for instance central facial palsy after stroke, and Parkinson’s Disease. Stroke occurs suddenly caused by disturbed blood flow, and oxygen deficiency (ischemic), or bleeding (hemorrhagic) in the brain leading to individual disabilities [37], whereas Parkinson’s Disease is a neurodegenerative disorder with loss of dopamine in the substantia nigra, resulting in typical symptoms of rigor, tremor, and bradykinesia [38]. Both, central facial palsy after stroke [26] [39] and Parkinson’s Disease [40] [41] [42] could result in similar effects, i.e. reduced facial expression, and therefore reduced facial feedback.
Following the FFH, facial feedback due to facial integrity is needed for facial emotion recognition [23]. Also, both in stroke [43], and Parkinson’s Disease [41], facial emotion recognition could be impaired. However, there is not necessarily a direct correlation between the limitations in facial expression and facial emotion recognition, at least in Parkinson’s Disease [41].”
Point 3: Regarding the study per se, I believe the results here reported are of great interest and maybe a comment in the discussion on the sidedness of the lesion (e..g., right vs left) and on its possible effect on both emotion expression and recognition would be of great interest. Did the authors investigate possible differences according to right or left facial palsy? Please, argue (See also: Ricciardi L, et al. Emotional facedness in Parkinson's disease. J Neural Transm (Vienna). 2018 Dec;125(12):1819-1827. doi: 10.1007/s00702-018-1945-6; and Marsili L, et al. Unraveling the asymmetry of Mona Lisa smile. Cortex. 2019 Nov;120:607-610. doi: 10.1016/j.cortex.2019.03.020).
Response 3: Thank you for your comment and your further interest. We studied all your recommended literature carefully.
We added new statistical analysis on the differences of facial emotion recognition according to left vs. right sided facial paresis in the Results, lines 353-358. We conducted t-tests to compare the accuracy of facial emotion recognition between patients with left vs. right sided facial paresis with
no significant difference. We run another t-test to compare the time of facial emotion recognition between patients with left vs. right sided facial paresis with no significant difference, also, lines 354-359:
“Moreover, a one tailed t-test for independent samples for facial emotion recognition in accuracy showed no significant difference in patients with left sided facial paresis (Mean=26.44±11.49) and right sided facial paresis (Mean=29.25±10.69), t(32)=-0.734; p=0.234. Another one tailed t-test for
independent samples for facial emotion recognition in time showed no significant difference in patients with left sided facial paresis (Mean=3.12±0.48) and right sided facial paresis (Mean=3.17±0.47), t(32)=-0.322; p=0.375.”
To guide the reader trough the manuscript, we add new information and literature to the topic facial emotion recognition depending on hemisphere and affected side of the face into the Introduction, lines 62-65:
“The localisation of emotion processing is also controversy discussed with evidence for right, left, or left and right hemispheric activation [19]. Dominance of the right hemisphere has been described historically [20], whereas recent evidence highlighted a combination of different neuronal networks with different lateralisation [19].”
and lines 94-96:
“More specifically, Korb et al. [29] reported differences depending on the paralysed side of the face with higher affected facial emotion recognition for patients with left sided than right sided facial palsy.”
And we discussed the topics of sidedness of the lesion as well as side of facial palsy in the section Discussion, Limitations of the study and added recommended literature, lines 461-463:
“Also, statistical adjustment for different stroke locations and lesion sizes would be beneficial, because differences in emotion recognition could depending on affected hemisphere [43].”
and lines 482-492:
“No significant differences were detected in objective facial emotion recognition accuracy, and time between patients with left or right sided facial paresis. If one hemisphere is dominant for emotion processing, [43] patients with lesion in this dominant hemisphere with contralateral facial paresis [25], could be possibly more affected. We cannot confirm this hypothesis and previous research in
facial palsy, where patients with left sided facial palsy showed less performance in facial emotion recognition compared to patient with right sided facial palsy [29]. But our results are in line with findings on patients with Parkinson’s Disease, where facial asymmetry is not related to hemispheric dominance for emotion processing [77]. For this, further evidence is needed to inspect possible differences in facial emotion recognition and expression depending on affected side of the facial palsy and hemisphere.”

This manuscript is a resubmission of an earlier submission. The following is a list of the peer review reports and author responses from that submission.
Round 1
Reviewer 1 Report
This manuscript investigates the ability to recognize visual and emotional expression in patients with stroke with facial palsy compared to stroke patients without facial palsy and compared to healthy controls. Their main results were that facial emotion recognition was impaired in stroke patients with facial palsy compared to stroke patients without facial palsy and also compared to controls. Furthermore, facial emotion recognition was worse than auditory emotion recognition in stroke patients with facial palsy.
Overall, the material presented in the manuscript is quite interesting. However, the manuscript is much too long and long-winded, thus, distracting the reader from key information on the present study by providing lot of detailed information not related to the study. It is recommended to shorten the manuscript significantly and to focus on the message that should be transported by the study.
The main shortcoming of the manuscript is that it completely neglects the heterogeneity of stroke symptoms and stroke localization. Emotion recognition is mainly a cortical function of the brain. Therefore, it is likely that strokes with large cortical involvement result in an impaired emotion recognition. Stroke with large cortical involvement are very likely accompanied with facial palsy. Strokes without facial palsy may be due to small cortical lesions, subcortical lacunar lesions, cerebellar lesions or brainstem lesions. To account for these differences, it would be necessary to have a detailed description of the brain areas affected by stroke and a statistical adjustment for different stroke locations and lesion sizes. Furthermore, it would be interesting whether facial palsy is the main
residual sequel of stroke or whether the patients had also other residual deficits, specifically aphasia or other neuro-cognitive deficits. It would be not surprising if the ability to recognize emotions are more affected by location and size of the stroke lesion than by the presence or absence of a facial palsy.
The presentation of the figures is insufficient. One would expect figures that show a set of boxplots for patients with facial palsy, for patients without facial palsy and also for controls. That means in total 3 boxplots per figure. It is not clear which statistical parameters the box and the whiskers represent (mean + standard deviation; median + interquartile range?)
If the group of stroke patients without facial palsy truly have no facial palsy, then the average Sunnybrook score should be 100. It is hard to understand, why the average score is only 90 in this group. It is further hard to understand how it could be possible that one third of those patients without facial palsy judge for themselves as having a facial palsy according to their own perspective.
Reviewer 2 Report
Review for Diagnostics
Reviewing “Is there a difference in Facial Emotion Recognition after Stroke with vs. without Central Facial Paresis? An observation study”
In this manuscript, the authors investigate the facial feedback hypothesis in two patient groups after stroke. One group (n = 34) with central facial paresis after stroke and a control group (n = 29) without central facial paresis after stroke. All participants complete a facial and a vocal emotion recognition test. Compared to the control group, the authors find significantly reduced emotion recognition accuracy (but not speed speed) in the facial paresis group for the facial emotion recognition test, but not for the vocal test, which is interpreted as supporting the facial feedback hypothesis.
Although the study is very interesting, specifically due to the choice of stroke patients for the paresis and the control group and I highly appreciate the considerate limitations section of the authors, I believe beyond the mentioned limitations the study design has additional weaknesses that limit any interpretation to a point that the current study cannot advance our understanding of the facial feedback hypothesis. (Facial) emotion perception ability is heavily g-loaded (Hildebrandt et al., 2015; Olderbak et al., 2019; Schlegel et al., 2020; for the most accurate representation of this relation I recommend Andrea Hildebrandt's work as she is the only one who works with measurement models based on multiple tests, thus providing the most accurate estimates of this relation). That means that any emotion perception test does not purely measure specific emotion perception ability, but also (to some degree) general mental capacity. Consequently, any group difference hypothesis (including the one presented here) about deficits in emotion perception ability must also consider general mental capacity, because the latter might moderate the tested effect to a degree that a positive finding might only be due to difference in g and not differences in emotion perception ability. For an example, similar ideas were recently investigated regarding emotional ability deficits in psychopaths and here any emotional deficit was actually completely due to g deficits (S. Olderbak et al., 2021; S. G. Olderbak et al., 2018).
The same problem arises regarding this study, but in the manuscript it is not mentioned whether participants completed a strong and reliable test battery to assess g (I recommend a composition as in Hildebrandt et al., 2015, that consists of gf/WMC, object perception, and immediate and delayed memory). However, g deficits as alternative hypothesis seem very likely when considering the sample details. A stroke presumably has an effect on general mental ability. This is supported by the large differences of both stroke groups compared to the norming samples of the cognitive (emotion perception) tests used in the study. However, therapy and time can presumably help to recover from such deficits. The control group on average is tested much later post onset than the facial paresis group (see Table A3), so they had way more time to recover, which might have helped them to recover more in g and thus also be better in emotion perception – independent of the facial paresis. One might argue, that then there should also be a group difference in the auditory emotion perception test, however, with only single tests we cannot directly infer to constructs (which we could with multiple tests and latent factor), so test differences (such as in reliability or how they individually relate to g) might explain this discrepancy. Thus, I must conclude that without a good and reliable g measure that allows to control for possible g-differences between the group and then, incrementally testing whether there are emotion perception differences, this study is too weak to contribute to the literature on the facial feedback hypothesis. Consequently, I must unfortunately recommend to reject the manuscript. However, I hope that future studies will overcome this limitation and then provide the so far strongest test of the facial feedback hypothesis.
References
Hildebrandt, A., Sommer, W., Schacht, A., & Wilhelm, O. (2015). Perceiving and remembering emotional facial expressions—A basic facet of emotional intelligence. Intelligence, 50, 52–67.
Olderbak, S. G., Mokros, A., Nitschke, J., Habermeyer, E., & Wilhelm, O. (2018). Psychopathic men: Deficits in general mental ability, not emotion perception. Journal of Abnormal Psychology, 127(3), 294.
Olderbak, S., Geiger, M., Hauser, N. C., Mokros, A., & Wilhelm, O. (2021). Emotion expression abilities and psychopathy. Personality Disorders: Theory, Research, and Treatment. https://doi.org/10.1037/per0000444
Olderbak, S., Semmler, M., & Doebler, P. (2019). Four-branch model of ability emotional intelligence with fluid and crystallized intelligence: A meta-analysis of relations. Emotion Review, 11(2), 166–183.
Schlegel, K., Palese, T., Mast, M. S., Rammsayer, T. H., Hall, J. A., & Murphy, N. A. (2020). A meta-analysis of the relationship between emotion recognition ability and intelligence. Cognition and Emotion, 34(2), 329–351. https://doi.org/10.1080/02699931.2019.1632801